# EcDBS1R4, an Antimicrobial Peptide Effective against *Escherichia coli* with In Vitro Fusogenic Ability

**DOI:** 10.3390/ijms21239104

**Published:** 2020-11-30

**Authors:** Marcin Makowski, Mário R. Felício, Isabel C. M. Fensterseifer, Octávio L. Franco, Nuno C. Santos, Sónia Gonçalves

**Affiliations:** 1Instituto de Medicina Molecular, Faculdade de Medicina, Universidade de Lisboa, 1649-028 Lisbon, Portugal; marcin.makowski@medicina.ulisboa.pt (M.M.); mrfelicio@medicina.ulisboa.pt (M.R.F.); 2Centro de Análises Proteômicas e Bioquímicas, Pós-graduação em Ciências Genômicas e Biotecnologia, Universidade Católica de Brasília, Brasília 71966-700, Brazil; bel.hidrox@gmail.com (I.C.M.F.); ocfranco@gmail.com (O.L.F.); 3S-Inova Biotech, Pós-graduação em Biotecnologia, Universidade Católica Dom Bosco, Campo Grande 79117-010, Brazil

**Keywords:** antimicrobial peptide, EcDBS1R4, cardiolipin, Gram-negative bacteria, *Escherichia coli*, hemifusion, hyperpolarization

## Abstract

Discovering antibiotic molecules able to hold the growing spread of antimicrobial resistance is one of the most urgent endeavors that public health must tackle. The case of Gram-negative bacterial pathogens is of special concern, as they are intrinsically resistant to many antibiotics, due to an outer membrane that constitutes an effective permeability barrier. Antimicrobial peptides (AMPs) have been pointed out as potential alternatives to conventional antibiotics, as their main mechanism of action is membrane disruption, arguably less prone to elicit resistance in pathogens. Here, we investigate the in vitro activity and selectivity of EcDBS1R4, a bioinspired AMP. To this purpose, we have used bacterial cells and model membrane systems mimicking both the inner and the outer membranes of *Escherichia coli*, and a variety of optical spectroscopic methodologies. EcDBS1R4 is effective against the Gram-negative *E. coli*, ineffective against the Gram-positive *Staphylococcus aureus* and noncytotoxic for human cells. EcDBS1R4 does not form stable pores in *E. coli*, as the peptide does not dissipate its membrane potential, suggesting an unusual mechanism of action. Interestingly, EcDBS1R4 promotes a hemi-fusion of vesicles mimicking the inner membrane of *E. coli*. This fusogenic ability of EcDBS1R4 requires the presence of phospholipids with a negative curvature and a negative charge. This finding suggests that EcDBS1R4 promotes a large lipid spatial reorganization able to reshape membrane curvature, with interesting biological implications herein discussed.

## 1. Introduction

The worldwide spread of antimicrobial resistance is among the biggest threats to public health [1]. However, pharmaceutical companies are leaving the antibiotic discovery research area due to the immense costs associated [2]. As a consequence, the pipeline of new antibiotics is suffering a long drought [3]. This situation is especially worrisome in the case of multidrug-resistant Gram-negative bacterial pathogens [4,5,6]. Gram-negative bacteria have an extra outer membrane surrounding its cells, conferring a permeability barrier against most hydrophobic molecules, turning them intrinsically resistant to several antibiotics [7,8]. Hence, the development of new strategies to eradicate Gram-negative bacteria without readily eliciting resistance has become a priority [9]. In this regard, antimicrobial peptides (AMPs) are considered promising candidates [10]. These peptides are ubiquitous in nature [11], characterized by an amphipathic character and a high a proportion of cationic amino acid residues [12]. This feature is key in the selectivity of AMPs, preventing them from binding to the closer-to-neutral surfaces of mammalian cells, while favoring the interaction with the anionic surfaces of bacteria [13]. The negative charge of Gram-negative bacterial membranes is conferred mostly by the lipid constituents. In Gram-negative organisms such as *Escherichia coli*, the highly anionic glycolipid lipopolysaccharide (LPS) is the major component of the outer leaflet of the outer membrane (OM). LPS is comprised of three regions: lipid A, a phosphorylated glucosamine disaccharide with four to six fatty acids, a core made up of a collection of branched oligosaccharides, and the O-antigen formed by repetitive monosaccharide subunits [14]. Most of the negative charges of LPS are provided by phosphate residues of lipid A [15,16], but also certain modifications in the core [17] and O-antigen domains [18]. The negative charges of LPS are stabilized by divalent cations such as Mg^2+^, providing the outer leaflet of the OM as a rigid structure, hard to permeate by hydrophilic or hydrophobic molecules [19]. The inner leaflet of the OM and the inner membrane (IM) have similar compositions, with proportions close to 60–70% of the zwitterionic phosphatidylethanolamine (PE), and the remaining 30–40% of the anionic phosphatidylglycerol (PG, 30–35%) and cardiolipin (CL, 5%), the last having two negative charges per molecule [20]. Most AMPs exert their antibacterial activity through the permeabilization of the cellular membranes, through mechanisms that vary depending on the properties of the peptide and the components of bacterial membranes [21].

In this work, we explored the activity and selectivity of the α-helical peptide EcDBS1R4 (PMKKKLAARILAKIVAPVW), recently designed using the linguistic model-based algorithm *Joker* [22]. EcDBS1R4 is not cytotoxic against human cells and is highly selective towards *E. coli*, not being effective against the Gram-positive pathogen *Staphylococcus aureus*. Flow cytometry assays indicate that EcDBS1R4 acts rapidly, killing 80% of cells within 15 min. Membrane selectivity was biophysically studied using biomembrane model systems, including models for the outer and inner membranes of *E. coli*, as well as *E. coli* cells, to validate vesicle studies. Our results suggest that binding to the LPS localized in the OM of *E. coli* is determinant for selectivity, while the antimicrobial activity is exerted through inner membrane disruption. The inability of the peptide to dissipate the electrochemical gradient in *E. coli* suggests that the antibacterial activity of the peptide is not exerted through the formation of stable pores. Moreover, we have found that EcDBS1R4 promotes the hemifusion of inner membrane-like (IML) vesicles. This result suggests that EcDBS1R4 induces major rearrangements in lipid distribution, resulting in the large curvature deformations necessary for hemifusion. Overall, our results have led us to propose a model for the mechanism of action of EcDBS1R4. Using artificial membranes mimicking those from bacteria, together with assays carried out with bacteria, we were able to get relevant insights into the mechanism of action of this AMP.

## 2. Results and Discussion

### 2.1. EcDBS1R4 Kills E. coli Rapidly and Is Non-Cytotoxic for Human Cells

We made a screening of the antimicrobial potential of EcDBS1R4. We tested the minimum inhibitory concentrations (MIC) of the peptide against different bacteria strains and compared its effectivity with three different conventional antibiotics (Table 1). EcDBS1R4 is effective against *E. coli* (ATCC 25922), with a MIC of 11.7 µM. EcDBS1R4 showed a slightly lower efficacy against the carbapenem-resistant *E. coli* strain 3789319. Similarly, in a recent study, this *E. coli* strain showed an increased resistance to the lipopeptide pelgipeptin A, when compared to a control strain typically used for antimicrobial testing [23]. Strikingly, this carbapenem resistant *E. coli* strain showed a 30-fold increased resistance to gentamicin when compared to *E. coli* ATTC 52922.

Interestingly, EcDBS1R4 was ineffective against the multi-drug resistant *K. pneumoniae* LACEN 3259271. *Klebsiella* and *Escherichia* are closely related genus of Gammaproteobacteria with similar membrane compositions [15,24]. There are several potential explanations for this seemingly counterintuitive observation, as Gram-negative bacteria can display a variety of defense mechanisms against AMPs. For instance, it has been extensively reported that the anionic capsule polysaccharide frequently coating the outer membrane of *K. pneumoniae* can act as a buffer against AMPs [25,26]. Another potential reason could be related to modifications in the structure of LPS [27]. We note that *E. coli* strains with similar adaptations have been described [28,29], and could therefore limit the effectiveness of EcDBS1R4 against such strains. Thus, future efforts aiming at identifying molecular players underneath the observed selectivity differences of EcDBS1R4 between closely related Gram-negative pathogens are needed.

EcDBS1R4 has also negligible antibacterial activity against the Gram-positive bacterium *S. aureus* (ATCC 25923). Three main lipid species make up the membrane of *S. aureus*: the anionic phosphatidylglycerol, the also anionic cardiolipin and variable proportions of lysyl-phosphatidyl glycerol (LPG) [30]. The lysine group of LPG confers it a net positive charge, and this feature has been linked to the resistance of some *S. aureus* strains to certain AMPs [30,31].

In contrast, in the conditions tested, EcDBS1R4 has no hemolytic potential against erythrocytes, indicating the human safety of this peptide. To gain an insight into the kinetics of its killing activity, we followed bacterial viability at different peptide concentrations and different incubation times, using flow cytometry (Figure 1). We found that bacterial lifetime is reduced at larger peptide concentrations. We estimated the time necessary to achieve 50% of compromised cells (t_1/2_) for all concentrations [32], and found that a 10 min exposure to 6 µM EcDBS1R4 is enough for kinetic stabilization of the antimicrobial activity (Figure 1B).

### 2.2. EcDBS1R4 Undergoes Conformational Changes Upon Membrane Interaction

We evaluated the structural changes of EcDBS1R4 in different environments, as they may have a major impact on the mode of action of AMPs [33]. Homology studies suggest that the preferential secondary structure is α-helical (Figure 2A), with most of the cationic charges concentrated in the N-terminal region (Figure 2B). We studied the peptide structure in different membrane environments, using circular dichroism (CD) and large unilamellar vesicles (LUVs) as biomembrane model systems. The chosen membrane models were the zwitterionic pure 1-palmitoyl-2-oleoyl-phosphatidylcholine (POPC), a mammalian membrane-like mixture of POPC and cholesterol (POPC:Chol 70:30), an anionic bacteria-like membrane of POPC and 1-palmitoyl-2-oleoyl-phosphatidylglycerol (POPC:POPG 70:30) and compositions mimetizing the outer (OML) and inner (IML) membranes of *E. coli*. Both OML and IML have 1-palmitoyl-2-oleoyl-phosphatidylehtanolamine (POPE) as the zwitterionic major phospholipid, and variable proportions of CL and POPG, and OML also includes LPS. Detailed compositions are summarized in Table 2. CD spectra (Figure 2C) indicate that the peptide is in a random coil when dissolved in buffer or in the presence of zwitterionic lipid vesicles. In contrast, the addition of anionic LUVs favors the adoption of an α-helical secondary structure. The structure response of the peptide was shown to be different depending on the lipid composition of the vesicles. To better understand this structural shift after membrane interaction, we performed a lipid titration following the signature signal for the α-helix, which allowed us to estimate an apparent dissociation constant (*K*_d,app_, Figure 2D and Table 2), which was associated with the lipid concentration at which the peptide changes its conformation. Lower *K*_d,app_ was obtained for IML and POPC:POPG (70:30) vesicles. The results indicate that EcDBS1R4 remains disordered in the presence of OML vesicles.

### 2.3. EcDBS1R4 Has an Increased Affinity towards Bacterial-Like Anionic Membranes

In vitro activity studies showed that EcDBS1R4 is not cytotoxic, but effective against Gram-negative bacteria. To better grasp the peptide specificity, we studied the EcDBS1R4 partition to diverse membrane model systems (Figure 3A). To this purpose, we monitored variations in the fluorescence emission spectrum of the tryptophan (Trp) residue of EcDBS1R4 upon titration with LUVs (lipid compositions are presented in Table 2). The relationship between fluorescence intensity changes and membrane partitioning is translated in the parameters calculated: the partition coefficient, *K*_p_ (i.e., the ratio of peptide concentrations between the membrane and the aqueous bulk), and *I*_L_/*I*_W_, the ratio between the fluorescence intensities with all the peptide in the membrane or in the aqueous environment (Table 2). Shifts in the Trp emission spectra were also recorded, as shifts towards more energetic wavelengths (blue shifts) are usually indicative of a transition of Trp into less polar environments (e.g., the hydrophobic core of lipid bilayers) [34]. Membrane incorporation of EcDBS1R4 varies depending on the lipid composition, being higher for the most anionic vesicles, and particularly high in the case of the IML vesicles, with a consistent 5 nm spectral blue shift. As for OML vesicles, although we were unable to determine a *K*_p_, a ~8.5 nm blue shift suggests that the membrane insertion of the Trp residue indeed occurs. Conversely, EcDBS1R4 displayed low quantum yield changes and no blue shift in the presence of zwitterionic vesicles (pure POPC and POPC:Chol), indicative of a negligible partitioning of the peptide into these membranes. The poor partitioning of the peptide into these zwitterionic model membranes provides solid insights into the reason behind the lack of cytotoxicity of EcDBS1R4 to human cells.

We further explored the membrane incorporation of EcDBS1R4, performing studies using acrylamide as a water-soluble quencher of the fluorescence of the Trp residue of EcDBS1R4, in the absence and presence of model membranes. Stern-Volmer plots obtained for EcDBS1R4 are shown in Figure 3B. Calculated Stern-Volmer constants (*K*_SV_) are summarized in Table 2. The *K*_SV_ obtained in the absence of LUVs (34.8 ± 0.43 M^−1^) was used as a control of no internalization. Addition of zwitterionic vesicles resulted in values close to the negative control, indicating negligible interaction with membranes. In contrast, *K*_SV_ values obtained with anionic lipid vesicles were significantly lower, implying that the Trp residue inserts into lipid bilayers of these compositions, in agreement with the partition experiment results. We obtained remarkably low values for IML and OML vesicles (9.7 and 13.1 M^−1^, respectively), aiding to better understand the membrane selectivity of the peptide (Table 2).

We performed additional experiments to study the selectivity of the peptide using the membrane dipole-sensitive probe di-8-ANEPPS, using *E. coli* cells as well as LUVs. Changes in the dipole potential caused by the interaction of the peptide are translated in alterations of the excitation spectrum. These changes, in turn, can be related to variations in affinity by means of the calculation of an apparent dissociation constant, *K*_d,app_, using Equation (6) [35], thereby enabling comparison of the affinities in both vesicles and cells. Figure 3C shows the variation in the R ratio in the different vesicles and in *E. coli* cells as a function of the peptide concentration. Parameters obtained from the fitting using Equation (6) (*K*_d,app_ and ΔR) are shown in Table 2. Peptide addition did not affect the dipole potential of zwitterionic vesicles, but altered that of anionic vesicles and *E. coli* cells. Higher *K*_d,app_, as for POPC:POPG (70:30) or IML vesicles, imply lower membrane affinities, when compared to the lower *K*_d,app_ determined for OML and *E. coli* cells (approximately 5.9 and 1.7 µM, respectively). As the common differential feature of OML vesicles and *E. coli* cells with respect to the other LUV compositions is the presence of LPS, we conclude that this lipid, by contributing to a largely anionic outer membrane, is a key factor in the affinity of EcDBS1R4 to *E. coli* cells. We note that the LPS used in OML vesicles is a rough-type LPS (O26:B6), whereas *E. coli* ATCC 25922 has a smooth–type LPS. Rough-type LPS frequently lack the O-antigen (the outermost component of the LPS molecule, composed by glycan polymers) [36]. These structural differences between rough- and smooth-type LPS may impact the binding of antimicrobial peptides. In fact, it has been suggested that rough LPS phenotypes are more susceptible to being targeted by AMPs than smooth LPS phenotypes [37]. The low dissociation constants displayed by EcDBS1R4 to *E. coli* ATTC 25992 suggest that the O-antigen length of LPS does not substantially impact the binding affinity of this peptide.

Complementary to these studies, we performed in silico molecular docking studies (Figure 3D), in membranes composed by POPC, POPC:Chol (70:30) and POPC:POPG (70:30). Simulations further suggest that EcDBS1R4 tends to interact more with the negatively charged membrane. Moreover, molecular docking studies suggest that the presence of Chol in membranes provides a molecular shield against EcDBS1R4. Evidences of a role of Chol in protecting eukaryotic membranes from the attack of AMPs are longstanding [38,39]. It is well established that Chol is critical for maintaining a mechanical stability of eukaryotic membranes, as Chol depletion causes increase in lipid bilayer fluidity and packing in cells [40,41]. The inter-atom interactions that occur below 3.6 Å are summarized in Appendix A. It should be noted, however, that docking simulations were performed using a single peptide molecule. Collective behaviors such as local oligomerization or aggregation at the membrane level could strongly affect membrane docking, limiting the extent of interpretation [42].

Overall, selectivity studies corroborate that EcDBS1R4 has an increased selectivity towards the bacterial or bacterial-like anionic membranes, and a negligible affinity to zwitterionic systems, a feature that aids to understand its lack of cytotoxicity against human cells. As for the negatively charged vesicles, di-8-ANEPPS studies suggest that the highly anionic LPS of the outer membrane of *E. coli* is a strong affinity determinant.

### 2.4. EcDBS1R4 Hyperpolarizes the Membrane Potential of E. coli and Increases the Packing of Inner Membrane-Like Vesicles

Aiming at gaining insights into the membrane activity of EcDBS1R4 in bacteria, we studied the effect of the peptide on the bacterial membrane potential and membrane packing, profiting from two fluorophores sensitivity to such membrane properties. Membrane potential was studied using DiOC_2_(3), a green fluorescent probe that undergoes a red shift in its fluorescence emission upon binding to polarized membranes [43]. Remarkably, EcDBS1R4 caused hyperpolarization of *E. coli* membranes (Figure 4A and Appendix A). This effect has been reported for other AMPs, such as the human granulysin or the synthetic Bac8c [44,45]. Importantly, the inability to dissipate membrane potential indicates that the antimicrobial activity of EcDBS1R4 is not exerted through the formation of stable pores, since such pores would dissipate the potential, contrary to what we observe [46,47].

Considering the result of membrane hyperpolarization, we were interested in assessing the packing properties of such membranes. We used the fluorescent probe Laurdan to measure the lipid packing of both vesicles and *E. coli* cells. The spectral emission of Laurdan is sensitive to local membrane hydration levels, and the generalized polarization (*GP*; Equation (7)) relates quantitatively these spectral changes. Figure 4B shows *GP* values of LUVs with different lipid compositions and *E. coli* cells as a function of EcDBS1R4 concentration. The interaction of the peptide caused little or no effect in zwitterionic vesicles. The highly packed OML vesicles remained unaffected as well. Conversely, POPC:POPG (70:30) and IML vesicles experience a slight increase in lipid packing upon addition of the peptide. Regarding *E. coli* cells, the results have high dispersion, but suggest that the peptide increases bacterial membrane packing. A pertinent question is whether Laurdan labels the outer, the inner or both membranes of *E. coli*. Previous reports [48] proposed that Laurdan inserts preferentially into the inner membrane of *E. coli*, rather than to the LPS-rich outer membrane. On this basis, changes in Laurdan fluorescence in *E. coli* would report the permeation of the peptide into the periplasmic space and the subsequent interaction with the inner membrane. This interpretation is consistent with our results, as the peptide changed the packing of IML vesicles, without doing so for the OML composition. Furthermore, the *GP* close to 0 found in *E. coli* cells indicates that the packing of the membranes is relatively low, when compared to the *GP* close to 0.22 for LPS-containing OML vesicles. This observation supports the notion that Laurdan localizes in the plasmatic membrane rather than in the LPS-rich, extremely packed outer membrane.

### 2.5. EcDBS1R4 Promotes Hemifusion of Membrane Mimics of the Inner Membrane of E. coli

Peptide-induced variations in membrane surface potential and membrane aggregation were studied using zeta-potential and dynamic light scattering (DLS) (Figure 4C,D, respectively) [43]. EcDBS1R4 promotes the aggregation of IML vesicles, without affecting the size distribution of any of the other membrane models studied. However, zeta-potential studies with LUVs and *E. coli* cells showed that EcDBS1R4 was unable to neutralize the surface charge of any of the vesicles studied here. It is worthy of notice that the aggregation of IML vesicles is not concomitant with an expected neutralization of the surface charge. Intriguingly, we found an overlap in the peptide concentration at which vesicle aggregation occurs and the stabilization of surface charge, settling at approximately 22 mV. As equally charged particles tend to repulse each other, we hypothesized that this puzzling aggregation could be explained by the fusion of IML vesicles promoted by EcDBS1R4.

To test this hypothesis, we performed lipid mixing assays, monitoring the Förster resonance energy transfer (FRET) efficiency between two lipid probes, using IML and zwitterionic vesicles, with increasing concentrations of peptide (Figure 4E). Indeed, IML vesicles were shown to be sensitive to peptide addition, with a maximum FRET efficiency of approximately 30% at 30 µM EcDBS1R4 (peptide-to-lipid ratio of 0.15).

The lipid composition of IML vesicles (PE:PG:CL63:33:4) may give us some hints about their peptide-driven hemifusion. CL is a phospholipid unique to energy transducing membranes (bacteria, mitochondrial inner membrane and chloroplasts), interacting intimately with the electron transport chain proteins. Two particularities characterize this phospholipid: a net charge of −2 at physiological pH and a small headgroup relative to the volume occupied by its four acyl chain tails [49]. The latter feature confers CL a high negative intrinsic curvature (or inverted cone-shaped). Therefore, local high concentrations of CL can strongly influence the membrane topography [49,50]. Phosphatidylethanolamine (PE), the zwitterionic main component of most bacterial membranes [51], has also a negative intrinsic curvature. It has been extensively reported that cationic amino acid residues can interact with CL, reducing the lipid–lipid electrostatic repulsion, and thereby promoting the formation of CL domains [52,53]. Thus, we propose that the peptide-driven hemifusion of IML vesicles might occur through the formation of CL-enriched nanodomains that would locally deform the membranes through curvature formation [54]. Subsequently, the warped outer leaflets of adjacent IML vesicles would form a concave stalk, leading to hemifusion [55]. It should be noted that the mechanism showed by EcDBS1R4 does not match the anionic clustering mechanistic paradigm [56], as EcDBS1R4 also promotes the fusion of vesicles lacking zwitterionic phospholipids (Appendix A). Indeed, we identify two main constraints for the fusogenic activity of EcDBS1R4, one geometric and the other of electrostatic nature. The geometric requirement is the presence of lipids with negative spontaneous curvature (conic-shaped lipids that favor the adoption of non-lamellar phases). The electrostatic condition is the presence of anionic lipids. Both requirements are met by CL, with its large negative intrinsic curvature and its two negative charges per molecule. It seems that in IML vesicles a specific interaction of EcDBS1R4 with CL is responsible for membrane wrapping that results in hemifusion process.

It is our understanding that vesicle hemifusion could be indicative of interactions with biological significance, as spatial rearrangements of CL may affect the functionality of key cell processes. CL contributes to the membrane structural stability, interacting intimately with several membrane proteins, namely with the respiratory chain complexes [57]. Among these, ATP synthase is a key regulator of the transmembrane potential required for cell viability [58]. Indeed, it has been speculated that CL could serve as a mechanical stabilizer and a lubricant for the rotor of the ATP synthase [59,60]. Thus, by hindering the necessary lipid interaction with essential proteins, EcDBS1R4 could be exerting an indirect toxic effect. This idea is supported by our observation of EcDBS1R4 promoting hyperpolarization of *E. coli* cells [61]. Peptide sequestration of CL molecules might affect the functionality of the respiratory chain [62], hampering the ability to dissipate the proton potential used for the phosphorylation of ADP into ATP, and thereby causing hyperpolarization of the cell membrane [63]. This hypothesis is under current exploration, and experiments suggest that it is a plausible possibility. A similar behavior was described for the cyclic peptide cWFW, which does not permeabilize bacterial membranes, killing by the formation of CL domains [64,65]. It is also worth of mention that CL has been also reported to be necessary for the function of proteins related to the infectiveness and drug resistance of enteric variants of the *Escherichia* and *Salmonella* genus, such as the PhoPQ system [61]. Conversely, it has been suggested that mutations causing increased CL synthesis turn the Gram-positive *S. aureus* resistant to the lipopeptide daptomycin [66]. It is thus pertinent to question whether homologous mutants in *E. coli* could render this peptide inefficient, by providing a CL buffer in the membrane. In fact, *S. aureus* has CL membrane concentrations of up to 32% [67], much higher than the proportions typically found in the inner membrane of *E. coli* [68,69]. This CL buffer of *S. aureus* may provide an additional explanation for its immunity to the action of EcDBS1R4, as proteins requiring CL for their correct function will have more of this phospholipid available, similarly to what was previously proposed by Epand and Epand [52]. Furthermore, in vivo experiments with *E. coli* cardiolipin synthase mutants (∆*clsABC*) [70] are required to fully comprehend the role of CL in modulating the in vivo activity and selectivity of EcDBS1R4.

Interestingly, EcDBS1R4 is inefficient against the multi-resistant strain of *K. pneumoniae* LACEN 3259271. *K. pneumoniae* and *E. coli* are closely related Gamma proteobacteria with similar membrane lipid compositions [24]. This raises the question of whether alterations in non-lipid membrane constituents, especially polysaccharides, could affect the selectivity and efficacy of EcDBS1R4. Future work should focus on the non-lipidic bacterial features that condition the efficacy of EcDBS1R4 and other antimicrobial peptides. In this sense, Fleeman et al. [71] have recently fine-tuned the amino-acid sequence of a peptide previously ineffective against recalcitrant capsulated *K. pneumoniae* strains, turning it able to disrupt the polysaccharide capsule barrier.

The present work explores the membrane selectivity and activity of the synthetic bioinspired antimicrobial peptide EcDBS1R4. We found that EcDBS1R4 is unable to form stable pores in *E. coli*, as it does not depolarize its membrane potential (Figure 4A). EcDBS1R4 has affinity to Gram-negative bacteria outer membrane-mimicking vesicles (Figure 3), but it does not affect their packing, surface charge or size distribution (Figure 4B–D). On the other hand, EcDBS1R4 elicits inner membrane-like vesicle hemifusion, suggesting that the membrane disruption activity by EcDBS1R4 takes place at the inner membrane level (Figure 4E).

## 3. Materials and Methods

### 3.1. Materials

EcDBS1R4 (PMKKKLAARILAKIVAPVW, 95% of purity) was purchased from Peptide 2.0 (Chantilly, VA, USA). Peptide was prepared in filtered Milli-Q H_2_O for the measurements and stored at 4 °C, protected from light. POPC, POPG and POPE were obtained from Avanti Polar Lipids (Alabaster, AL, USA). Bovine heart cardiolipin sodium salt, cholesterol (Chol), lipopolysaccharide from *Escherichia coli* O26:B26 (LPS), Luria-Bertani Agar (LB), Mueller Hinton Broth (MHB), Pluronic F-127, dipalmitoyl-phosphatidylethanolamine-sulforhodamine B (Rh-PE) and 1,2-dihexadecanoyl-*sn*-glycero-3-phospho-[N-4-nitro-benz-2-oxa-1,3-diazolyl]-ethanolamine (NBD-PE) were purchased from Sigma Aldrich (St. Louis, MO, USA). The probes Laurdan (2-dimethylamino-6-lauroylnaphtalene), DiOC_2_(3) (3,3′-diethyloxacarbocyanine iodide and di-8-ANEPPS (4-[2-[6-(dioctylamino)-2-naphthalenyl]ethenyl]-1-(3-sulfopropyl)-pyridinium) were purchased from Invitrogen (Carlsbad, CA, USA). HEPES 10 mM, with NaCl 150 mM, pH 7.4, was used as working buffer.

### 3.2. Large Unilamellar Vesicles Preparation

Large unilamellar vesicles (LUVs) of approximately 100 nm of diameter were prepared by extrusion [72,73]. Lipids were dissolved in chloroform; the solvent was removed under a nitrogen steam and the lipid film was placed on a vacuum pump overnight. In the case of the composition containing lipopolysaccharide (LPS), film formation was obtained dissolving LPS in chloroform:methanol (2:1), and the solution was vortexed and bath-sonicated at 40 °C for 15 min [73]. The resulting lipid films were then rehydrated with buffer, frozen and thawed, and extruded through a membrane with a pore size of 100 nm (Whatman, Florham Park, NJ, USA), using a LiposoFast-Basic extruder (Avestin Europe, Mannheim, Germany)^56^. Information on the compositions used are shown in Table 2.

### 3.3. Bacteria Cells Preparation

An aliquot of *E. coli* ATCC 25922 (ATTC, Manassas, VA, USA) was plated on LB agar, and incubated overnight at 37 °C. An isolated colony was re-suspended into 5 mL of MHB and grown overnight at 37 °C, until the log-phase state was reached. A 100 µL inoculate was transferred into 5 mL of fresh medium and grown for 3 h, then centrifuged at 4000× *g*, for 25 min, at 10 °C, and washed three times with MHB medium. The optical density at 600 nm of the washed cells (OD_600_) was measured and adjusted to fit the experimental conditions.

### 3.4. In Silico Analysis

#### 3.4.1. Molecular Modelling

The structure model template of EcDBS1R4 was constructed by molecular modelling using protein–protein BLAST, as described elsewhere [74]. The Protein Data Bank structure used as a template was 1WFA, and the three-dimensional structure of EcDBS1R4 was constructed using Modeller v. 9.12. The best model was chosen based on its geometry, stereochemistry, and energy distribution, using PROCHECK. 3DSS was also used to calculate the root mean square deviation [75].

#### 3.4.2. Molecular Modelling

Preferential binding interactions between the peptide and anionic or zwitterionic membranes were studied by molecular docking using AUTODOCK v. 4.2, as previously described [74]. The CHARMM-GUI server was used for membrane construction similar to the lipid vesicles prepared in vitro, with 75 × 75 × 75 Å^3^ of dimension. The grid box was calculated with 35 × 35 × 15 points and 1 Å of spacing centered on the membrane surface. Maximum side chain flexibility was enabled. Molecular docking simulations were programmed for 50 random runs, and the obtained structures were ranked according to their affinity values. PyMOL was used to predict peptide–membrane interactions, an applied distance cutoff of 3.6 Å.

### 3.5. In Vitro Activity Assays

#### 3.5.1. Minimum Inhibitory Concentration Determinations

Minimum inhibitory concentrations (MIC) of peptides against *E. coli* ATCC 25922, *S. aureus* ATCC 25923, and two multidrug resistant strains obtained from Laboratório Central of Brasília (LACEN), the carbapenemase-producing *E. coli* (LACEN 3789319) and *K. pneumoniae* (LACEN 3259271), isolated from the blood from hospitalized patients, were determined using the standardized dilution method, according to Clinical and Laboratory Standards Institute guidelines [76]. Experiments were performed in triplicate.

#### 3.5.2. Hemolysis Assays

Hemolytic activity was evaluated against fresh C57BL/6 mice erythrocytes collected in EDTA vacuum collection tubes, as previously described [77]. Briefly, 50 µL of EcDBS1R4 (200–2 µg/mL) were mixed with 50 µL of erythrocyte suspension (1% in PBS). Triton X-100 (0.1% *v/v*) and PBS were used as positive and negative controls, respectively. Hemolysis was measured at 540 nm, using a Powerwave HV microplate reader (BioTek, Winooski, VT, USA).

### 3.6. Flow Cytometry

Data were acquired in a BD Accuri C6 Flow Cytometer (BD Biosciences, San Jose, CA, USA), using blue and red excitation lasers (488 nm and 640 nm, respectively). Green and red fluorescence emission were detected with 530 and 670 nm bandpass filters, respectively. Fluorescence emission was acquired in a biexponential scale, collecting 10,000 events. Assays were conducted in triplicates, at room temperature. Flow cytometry results were analyzed using FlowJo Software v. 10.0× (Tree Star, Ashland, OR, USA).

#### 3.6.1. Cell Viability

For the determination of the percentage of live/dead bacteria after incubation with peptide, flow cytometry analysis was performed using the live/dead kit assay (Life Technologies, Carlsbad, CA, USA), as described elsewhere [78]. Time necessary to compromise 50% of the bacteria population was calculated as previously described [32].

#### 3.6.2. Membrane Potential

Membrane potential alterations were determined using the DiOC_2_(3) dye, as previously described [43]. *E. coli* cells were incubated for 1 h with EcDBS1R4 and stained with 15 µM of the dye. Cells were also incubated with 10 µM of carbonyl cyanide 3-chlorophenylhydrazone (CCCP), as a positive control. The red/green fluorescence ratio was calculated using the fluorescence intensities determined.

### 3.7. Circular Dichroism

CD measurements were carried out in a JASCO J-815 spectropolarimeter (Tokyo, Japan), using cuvettes of 0.5 cm path length. Spectra were acquired between 200 and 260 nm, at 25 °C, with a data pitch of 0.5 nm, a wavelength sampling velocity of 200 nm/min, and a data integration time of 1 s, performing at least 5 accumulations. Measurements were conducted with different lipid concentrations, up to 1.5 mM, in the absence or presence of 20 µM EcDBS1R4. In addition to blank subtraction, experimental instrument-related baseline drift was corrected by subtracting to all spectra the average of the signal between 250 and 260 nm. Spectra were normalized to mean residue molar ellipticity (*θ*, deg.cm^2^/dmol). All conditions were measured independently and in triplicate. Apparent affinity parameters were obtained by fitting using Equation (1) [79]:(1)$$\Delta \theta_{222\mathrm{nm}}=\frac{\Delta \theta_{222\mathrm{nm},\max}\;{\left[L\right]}^{n}}{{\left(K_{d,\mathrm{app}}\right)}^{n}+\;{\left[L\right]}^{n}}$$
with Δ*θ*_222nm_ being the difference of the peptide signal in the presence and absence (*θ*_0_) of lipid, Δ*θ*_222nm,max_ the maximum ellipticity value obtained from the fitting, [L] the lipid concentration, *n* the cooperativity constant and *K*_d,app_ the half-maximal effect, which represents the apparent dissociation constant. Error bars on data presentation represent the standard deviation of the triplicates.

### 3.8. Fluorescence Spectroscopy

Fluorescence measurements were carried out in a Varian Carry Eclipse fluorescence spectrophotometer (Mulgrave, Victoria, Australia) equipped with a Xe pulsed lamp. Excitation and emission bandwidths were 5 and 10 nm, respectively. Fluorescence spectra were recorded with 0.5 cm path length quartz cuvettes. All measurements were performed at least three times, with independent measurements, at 25 °C. All fluorescence spectra were corrected for scattering and dilution effects.

#### 3.8.1. Partition Coefficient Determination

To study peptide partition to different phospholipid bilayers, LUVs with the compositions summarized in Table 2 were used. Peptide partitioning was monitored by following the intrinsic fluorescence of the C-terminal tryptophan residue of the peptide. A titration of 6 µM EcDBS1R4 with LUVs was performed. After each vesicle addition, the sample was incubated for 5 min before recording the emission spectrum. Fluorescence emission spectra between 300 and 500 nm were recorded with excitation at 280 nm. Non-partitioning free Trp was titrated under the same conditions. To quantify the preference of the peptide for the membrane environment relative to the aqueous medium, the partition coefficient, *K*_p_, was calculated from the fit of Equation (2) to the experimental data [80]:(2)$$\frac{I}{I_{W}}=\frac{1+K_{p}\gamma_{L}\frac{I_{L}}{I_{W}}\;\left[L\right]}{1+K_{p}\gamma_{L}\;\left[L\right]}$$
where *I*_W_ and *I*_L_ are the fluorescence intensities with all the peptide in water and in the lipid phase, respectively, *γ*_L_ is the lipid molar volume and [L] is the lipid concentration. For non-hyperbolic partitioning behaviors, *K*_p_ was calculated using Equation (3) [81]:(3)$$\frac{I}{I_{W}}=\;\frac{K_{p}\gamma_{L}\;\left[L\right]I_{L}}{1+K_{p}\gamma_{L}\left[L\right]+k_{2}K_{p}I_{L}}+\;\frac{I_{w}}{1+K_{p}\gamma_{L}\;\left[L\right]}$$
where *k*_2_ is proportional to the ratio between the bimolecular self-quenching rate and the radiative decay rate.

#### 3.8.2. Acrylamide Quenching

Exposure of the Trp residue of the peptide to the aqueous environment was evaluated by fluorescence quenching with acrylamide and 6 µM of a peptide solution in the absence and presence of lipid vesicles (3 mM) were titrated with increasing concentrations of acrylamide. Trp emission was monitored between 300 and 500 nm, with excitation at 290 nm, to minimize the quencher/fluorophore light absorption ratios. The extent of linear quenching of the Trp residue was quantified using the Stern-Volmer Equation (4):(4)$$\frac{I_{0}}{I}=1+K_{SV}\;\left[Q\right]$$
where *I*_0_ is the fluorescence intensity of the peptide in the absence of quencher, *I* is the fluorescence intensity at a given quencher concentration, [Q], and *K*_SV_ the Stern-Volmer constant [34]. For negative deviations to a linear behavior on the Stern-Volmer plot (*I*_0_/*I* vs. [*Q*]), Equation (5) was used to fit the experimental data:(5)$$\frac{I_{0}}{I}=\frac{1\;+\;K_{SV}\;\left[Q\right]}{\left(1\;+\;K_{SV}\;\left[Q\right]\right)\left(1-f_{B}\right)\;+\;f_{B}}$$
where *f_B_* is the fraction of light arising from the fluorophores accessible to the quencher [82].

#### 3.8.3. Membrane Probes

*Di-8-ANEPPS:* Perturbations in the dipolar potential of membranes due to the interaction with the peptide were monitored by assessing the shift in the excitation spectra of di-8-ANEPPS [83]. LUVs were incubated at a probe-to-lipid ratio of 1:300 and a final lipid concentration of 3 mM. In the case of bacteria, a suspension of *E. coli* with a cell density of 1 × 10^4^ was labelled with a final dye concentration of 100 µM. For both LUVs and bacteria, the incubation time was 60 min. After labelling, peptide was added and incubated for 1 h. Excitation spectra were recorded as a function of peptide concentration, between 380 and 580 nm, with emission at 670 nm. The ratio of intensities at the excitation wavelengths of 455 and 525 nm (*R* = *I*_455_/*I*_525_) was calculated for the different peptide concentrations, and Equation (6) was used to fit these results [35]:(6)$$\frac{R}{R_{0}}=\;\frac{\frac{R_{\min}}{R_{0}}\;\left[P\right]}{K_{d}+\;\left[P\right]}$$
with [P] being the peptide concentration, *R*_0_ the intensity ratio without peptide, *R*_min_ the minimum asymptotic value for *R* and *K*_d,app_ the apparent dissociation constant.

*Laurdan:* Variations in lipid packing of bacterial cell membranes and LUVs upon addition of peptide were monitored by assessing the emission spectra shifts of Laurdan. Labelling procedure was essentially the same as for di-8-ANEPPS, except the dye concentration, which was 10 µM for Laurdan, and the incubation time, which was 30 min. Labelled samples were excited at 350 nm, and the emission between 400 and 600 nm was recorded. The emission of Laurdan is sensitive to the lipid bilayer phase state, and thus, to quantify the spectral changes, the Laurdan generalized polarization (*GP*) was calculated using Equation (7):(7)$$GP=\;\frac{I_{440}-\;I_{490}}{I_{440}+\;I_{490}}$$
where *I*_440_ and *I*_490_ are fluorescence intensities with emission at 440 and 490 nm, respectively [84].

#### 3.8.4. Hemifusion Efficiency Determinations

Peptide-induced lipid (hemi)fusion in vesicles was measured by Förster resonance energy transfer (FRET), using two probes (Rh-PE and NBD-PE), as previously described [85]. Briefly, unlabeled and double-labelled (containing 0.5 mol% of each fluorophore) vesicles obtained by extrusion were mixed at a 1:4 proportion, to a final lipid concentration of 200 µM. Fluorescence emission was measured between 475 and 650 nm, with excitation at 470 nm, after 10 min of peptide incubation. Fusion efficiency was quantified using Equation (8) [86]:(8)$$\%\;Fusion\;Efficiency\;=\;\frac{R-R_{0}}{R_{100\%}-R_{0}}$$
where *R* is the value of the ratio between the fluorescence intensities with emission at 530 and 588 nm, corresponding to the maximum fluorescence emission of NBD and RhB, respectively, *R*_0_ is the ratio before peptide addition (constant during the evaluated time range) and *R*_100%_ the ratio after addition of Triton X-100 at a final concentration of 1% (*v/v*).

### 3.9. Light Scattering Spectroscopy

Light scattering measurements were carried out in a Malvern Zetasizer Nano ZS (Malvern, UK), with backscattering detection at a constant angle of 173°, equipped with an He-Ne laser (λ = 632.8 nm). Samples were left equilibrating for 15 min at 25 °C. All samples were firstly filtered with nylon filters with 100 nm of diameter.

#### 3.9.1. Zeta-Potential Measurements

Zeta-potential measurements were performed using disposable folded capillary cells with golden electrodes. Successive aliquots of peptide were added into suspensions of either bacteria (1 × 10^6^ cells/mL) or LUVs (200 µM) diluted in filtered HEPES buffer and incubated 1 h before measurement. Values were obtained from an average of 15 measurements (100 runs each) [43].

#### 3.9.2. Dynamic Light Scattering Measurements

Changes in the size distribution of LUVs due to the addition of the peptide were determined by dynamic light scattering (DLS). Aliquots of EcDBS1R4 were added to a suspension of vesicles with a lipid concentration of 200 µM, diluted in HEPES buffer. The hydrodynamic diameter (*D_H_*) was obtained from 15 measurements (10 runs each).

## Figures and Tables

**Figure 1 ijms-21-09104-f001:**
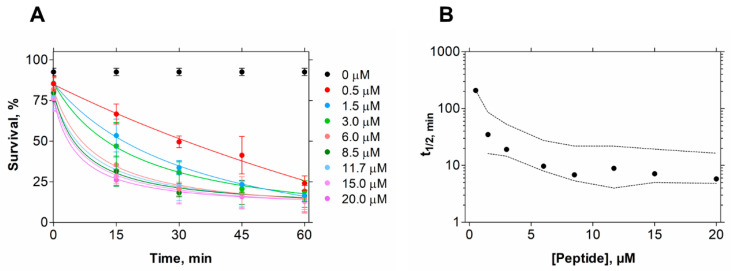
*E. coli* cell viability kinetics after EcDBS1R4 treatment. (**A**) Bacteria cell viability followed for 1 h by flow cytometry as a function of peptide concentration. (**B**) Time kinetics of cell viability after peptide treatment. Each point represents the time necessary to produce 50% of death for each concentration tested. Dashed lines indicate 95% confidence intervals.

**Figure 2 ijms-21-09104-f002:**
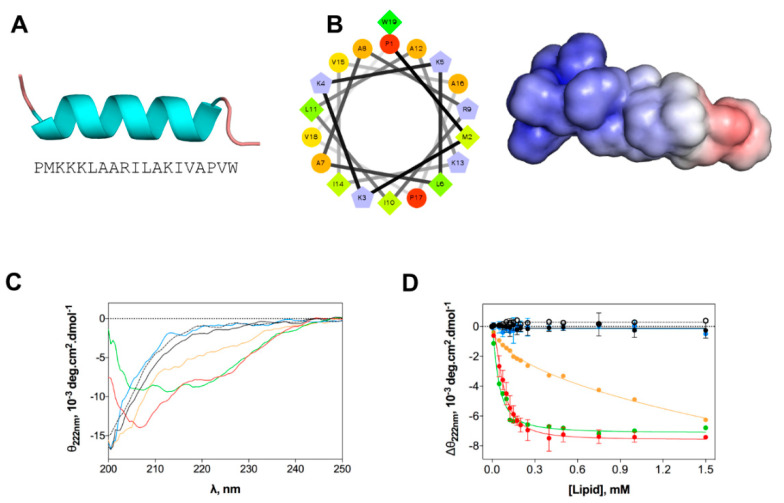
In silico and experimental structural studies of EcDBS1R4 structure. (**A**) Amino acid sequence of EcDBS1R4 and theoretical three-dimensional structure predicted by homology and constructed using Modeller v. 9.12. (**B**) α-helical wheel (constructed using the server http://rzlab.ucr.edu/scripts/wheel/wheel.cgi; hydrophobic residues are represented as green diamonds, hydrophilic, non-charged residues are coded as orange circles, and positively charged residues are blue-grey pentagons) and adaptive Poisson-Boltzmann solver electrostatic potential of EcDBS1R4, ranging from −5.0 k_B_T/e (red) to +5.0 k_B_T/e (blue). (**C**,**D**) Experimental structural studies of EcDBS1R4 in the presence of different LUV compositions. (**C**) Circular dichroism spectra of EcDBS1R4 in solution (dotted black) at 16 µM, and in the presence of 500 µM of lipid vesicles of POPC (black), POPC:Chol (70:30) (blue), POPC:POPG (70:30) (red), inner (green) and outer (orange) membrane of *E. coli* mimetic systems. (**D**) Comparative plot of the θ signal at 222 nm (a local minimum for α-helices) at different lipid concentrations, with the same lipid composition described above. Solid lines represent fits to the experimental data using Equation (1). Each experiment was conducted in triplicate and represented as mean ± standard deviation (SD). Calculated parameters are presented in Table 2.

**Figure 3 ijms-21-09104-f003:**
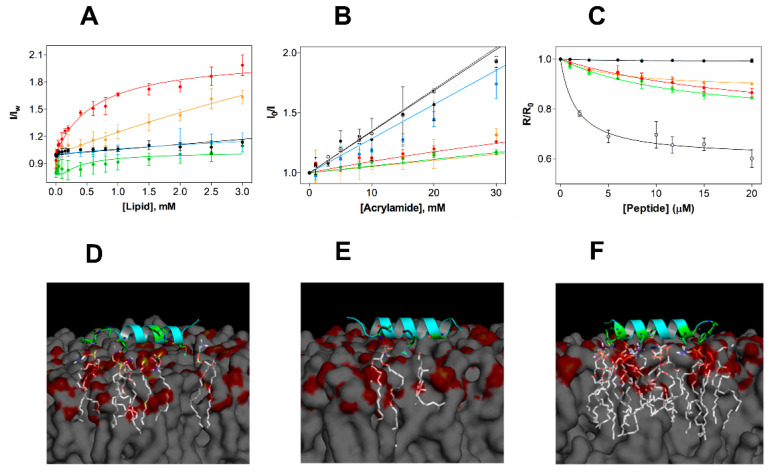
Experimental and in silico studies of the membrane selectivity of EcDBS1R4. (**A**) Partition curves of 6 µM EcDBS1R4 to LUVs of different compositions. Solid lines represent fits obtained using Equations (2) and (3). The LUV compositions used were POPC (black), POPC:Chol 70:30 (blue), POPC:POPG 70:30 (red), IML (POPE:POPG:CL 63:33:4) (green) and OML (POPE:POPG:CL:LPS 80:16:1:3) (orange). (**B**) Fluorescence quenching by acrylamide of 6 µM EcDBS1R4 in the absence (open circles) and presence of LUVs (color code as in A). Solid lines represent fits obtained using Equations (4) and (5). (**C**) Changes in membrane dipole potential as a function of EcDBS1R4 concentration, measured with di-8-ANEPPS (4-(2- [6-(dioctylamino)-2-naphthalenyl]ethenyl)-1-(3-sulfopropyl)pyridinium inner salt) in LUVs (200 µM lipid concentration; color code as in A) and *E. coli* cells (1 × 10^4^ cells/mL), represented by grey filled circles. The plot represents the di-8-ANEPPS excitation ratio *R* (*I*_455nm_/*I*_525nm_) for each peptide concentration, normalized divideing by *R*_0_ (the *R* value in the absence of peptide). Solid lines represent fits obtained using Equation (6). The parameters obtained from the fittings are summarized in Table 2. All experiments were conducted in triplicate. (**D**–**F**) Three-dimensional theoretical representation of the peptide interacting with membranes composed of POPC (**D**), POPC:Chol (70:30) (**E**) and POPC:POPG (70:30) (**F**), indicating the amino acid residues (in green) and the phospholipid molecules (in white) possibly involved in the interactions (residues involved and distances of interactions occurring for each membrane are detailed in Supporting Information Table 1).

**Figure 4 ijms-21-09104-f004:**
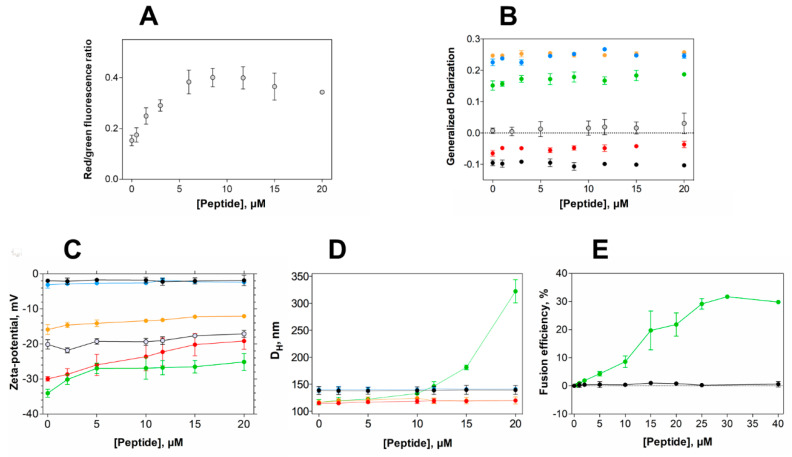
Alterations in membrane properties induced by EcDBS1R4. (**A**) Effect of EcDBS1R4 on the transmembrane potential of *E. coli* cells, calculated from the DiOC_2_(3) green shift. (**B**) Generalized polarization of LUVs (3 mM lipid) and *E. coli* cells (1 × 10^4^ cells/mL) labelled with Laurdan, as a function of peptide concentration. (**C**) Zeta-potential of LUVs (200 µM lipid) and *E. coli* cells as a function of peptide concentration. (**D**) Hydrodynamic diameter of LUVs (200 µM lipid), measured by dynamic light scattering as a function of EcDBS1R4 concentration. (**E**) Fusion/hemifusion efficiency of IML (PE:PG:CL (63:33:4)) LUVs (200 µM lipid) as a function of peptide concentration, calculated using Equation (8). The LUV compositions used were POPC (black), POPC:Chol 70:30 (blue), POPC:POPG 70:30 (red), IML (POPE:POPG:CL 63:33:4) (green) and OML (POPE:POPG:CL:LPS 80:16:1:3) (orange). *E. coli* cells are represented by grey open circles. All experiments were conducted in triplicate. Data are presented as mean ± SD.

**Table 1 ijms-21-09104-t001:** EcDBS1R4 minimal inhibitory concentrations (MIC) calculated for different bacteria strains, as well as erythrocyte hemolysis compared with commercial antibiotics.

	MIC								Hemolysis	
	*E. coli* (ATCC 25922)		*E. coli* (3789319)		*S. aureus* (ATCC 25923)		*K. pneumoniae* (LACEN3259271)		Erythrocytes	
	μM	μg/mL	μM	μg/mL	μM	μg/mL	μM	μg/mL	μM	μg/mL
EcDBS1R4	11.7	25	30	64	>120.1	>260	60	130	>93.8	>200
Gentamicin	<1.88	<1	60	29	<1.88	<1	0.23	0.1	-	-
Imipenem	120.1	36	>240.1	>72	15	4	>240.1	>72	-	-
Cefotaxime	<1.88	<1	>60	>27	<1.88	<1	7.5	3	-	-

**Table 2 ijms-21-09104-t002:** Membrane selectivity parameters obtained and details of the lipid compositions of the systems used.

Membrane Models Tested		CD		Partition			Quenching	Di-8-ANEPPS
Composition	Membrane Mimic	*K*_d,app_ (μM)	*n*	*K* _p_	*I*_L_/*I*_W_	Blue Shift (nm)	*K*_SV_ (M^−1^)	*K*_d,app_ (μM)
Free in solution	-	n.d. ^a^	n.d. ^a^	n.a.^b^	n.a. ^b^	n.a.^b^	34.8 ± 0.43	n.a. ^b^
POPC	Control	n.d. ^a^	n.d. ^a^	n.d.^a^	n.d. ^a^	0	34.2 ± 0.57	n.d. ^a^
POPC:Chol (70:30)	Mammalian cells	n.d. ^a^	n.d. ^a^	263 ± 158	1.4 ± 0.19	0	28.5 ± 1.1	n.d. ^a^
POPC:POPG (70:30)	General model of bacteria	75.0 ± 4.0	1.8 ± 0.19	2064 ± 292	2.1 ± 0.05	11	18.7 ± 4.2	15.2 ± 2.6
POPE:POPG:CL (63:33:4)	Inner membrane of *E.coli*	44.0 ± 4.0	1.4 ± 0.17	21615 ± 8407	1.0 ± 0.05	5	5.6 ± 0.14	13.4 ± 3.7
POPE:POPG:CL:LPS (80:16:1:3)	Outer membrane of *E. coli*	n.d. ^a^	n.d. ^a^	n.d ^a^	n.d ^a^	8.5	5.2 ± 0.28	9.6 ± 1.4
*E. coli* ATCC 25922	-	n.a. ^b^	n.a. ^b^	n.a. ^b^	n.a. ^b^	n.a. ^b^	n.a. ^b^	1.7 ± 0.49

n.d. ^a^—the equation could not fit the data, not possible to determine; n.a. ^b^—data not collected, not applicable.

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
