# Peer review of "EcDBS1R4, an Antimicrobial Peptide Effective against Escherichia coli with In Vitro Fusogenic Ability"

_ijms, 2020, doi:10.3390/ijms21239104_

Round 1

Reviewer 1 Report

The authors satisfactorily addressed all my comments. 

I found a couple of typos, such as E. coli not being italicized in pages 217 and 222. The authors should revise the text one more time.

Author Response

Reviewer 1:

Comments

The authors satisfactorily addressed all my comments. 

I found a couple of typos, such as E. coli not being italicized in pages 217 and 222. The authors should revise the text one more time.

Reply: Typos were corrected throughout the manuscript.

We thank Reviewer 1 for the comments and suggestion.

Reviewer 2 Report

While the manuscript is partly improved. There are few issues that authors need to address. The way it is written, it is clear that in vivo data is missing and needs serious qualification of various claims.

 1. To claim EcDBS1R4 has increased affinity for cardiolipin containing membranes needs to be qualified as authors do not show toxicity assay in E. coli strains lacking cls genes. If authors cannot perform such experiments, they should qualify their statements and mention that their results needs additional in vivo experiments with cls mutants.

2. Authors claim that Kleibesella is more resistant than E. coli due to lipid A modifications in the former needs again qualification. Authors must state at present they lack additional data or control experiments. Lipid A of E. coli also can be modified by P-ETN/L-Ara4N that are critical for resistance. Unless strains that constitutively express BasS/R or lack this two-component system, statement about lipid A modifications is not pertinent and needs to be qualified.

3. The reference no 14 cited in the Introduction section is not relevant as that review does not address what is claimed in lines 52-55. Authors should cite references that directly deal with LPS modifications and its structure.

Author Response

Reviewer 2:

While the manuscript is partly improved. There are few issues that authors need to address. The way it is written, it is clear that in vivo data is missing and needs serious qualification of various claims.

Comments:

  1. To claim EcDBS1R4 has increased affinity for cardiolipin containing membranes needs to be qualified as authors do not show toxicity assay in E. coli strains lacking cls genes. If authors cannot perform such experiments, they should qualify their statements and mention that their results needs additional in vivo experiments with cls mutants.

Reply: We would like to acknowledge Reviewer 2 for the careful reading and the recommendations, which undoubtedly will improve the quality of the manuscript. We agree that in vivo data would further support the claims of selectivity towards cardiolipin in bacterial membranes. Experiments at that level are part of our ongoing work, but those are far from being completed. As proposed, we have qualified our claims in this regard throughout the manuscript, highlighting the need for further in vivo experiments. In particular, we now understand that in the abstract this issue led to confusion by mixing in vivo selectivity results (effectivity against different bacteria) with in vitro experiments (using model membranes).

Line 28-29: The sentence “We found that EcDBS1R4 has an increased affinity towards membranes containing cardiolipin” was deleted.

Line 354-356: We added the following sentence: Further in vivo experiments with E. coli CL synthase mutants (∆clsABC) [71] will be necessary to fully comprehend the role of CL in modulating the activity and selectivity of EcDBS1R4.

  1. Authors claim that Klebsiellais more resistant than  coli due to lipid A modifications in the former needs again qualification. Authors must state at present they lack additional data or control experiments. Lipid A of E. coli also can be modified by P-ETN/L-Ara4N that are critical for resistance. Unless strains that constitutively express BasS/R or lack this two-component system, statement about lipid A modifications is not pertinent and needs to be qualified.

Reply: We agree with this observation. Accordingly, we have rephrased the paragraph to make the speculative tone of the claim clearer. In addition, we have clarified that the lipid A of E. coli can also be modified, stating that this can lead to resistance to AMPs. The molecular players that determine the effectivity of EcDBS1R4 in closely related Gram-negative pathogens should be further explored in the future.

Line 98-102: We have changed the previous sentence (“However, multi-drug resistant K. pneumoniae frequently undergo a lipid A remodeling process that renders them resistant to several AMPs”) to:

[…] Another potential reason [for the observed difference in activity between closely related bacteria K. pneumoniae and E. coli, with similar membrane compositions, the first being polysaccharide capsule frequently found in K. pneumoniae] could be related to modifications in the structure of LPS [29]. We note that E. coli strains with similar adaptations have been described [30,31] and could therefore limit the effectiveness of EcDBS1R4 against such strains. Thus, future efforts aiming at identifying molecular players underneath the observed selectivity differences of EcDBS1R4 between closely related Gram-negative pathogens are needed.

Line 361-362: Future work should focus on the non-lipidic bacterial features that condition the efficacy of EcDBS1R4 and other antimicrobial peptides.

  1. The reference no 14 cited in the Introduction section is not relevant as that review does not address what is claimed in lines 52-55. Authors should cite references that directly deal with LPS modifications and its structure

Reply: We have changed the references accordingly

Line 58-59: References 15-19

Round 2

Reviewer 2 Report

The revised manuscript by Makowski is vastly improved. However, several references to LPS structure and modifications  are misquoted and are not relevant to what is stated in text lines 54 to 58 and lines 94 to 100. Please replace citations as suggested below which are more appropriate.  

The references for LPS section are not authentic ones. It is best to use well cited papers by experts with relevant coverage. Remove references 14 to 18. Reference 19 is fine. Remove reference 25 not relevant. Also remove reference no 30 as it covers a totally different aspect.

Replace ref 14 by most widely used review (Raetz, C.R.H.; Whitfield, C. Lipopolysaccharide endotoxins. Annu. Rev. Biochem. 2002.

Replace ref 15 by a very good review by  Raetz, C.R.; Reynolds, C.M.; Trent, M. S.; Bishop, R.E. Lipid A modification systems in Gram-negative bacteria. Annu. Rev. Biochem. 2007, 76,

Replace Ref 16 by Klein, G et al J. Biol. Chem. 2011, 286,. This deals with lipid A and core alterations.

Replace ref 17 by Klein, G.; Müller-Loennies, S …. J. Biol. Chem. 2013, 288 describes phosphate charges and charge compensation by modification of core region.

You can cite review by Raetz et al   Annu. Rev. Biochem. 2007, 76 instead of no 25.

Replace ref 30 by B. D. Needham, and M. S. Trent, “Fortifying the barrier: the impact of lipid A remodelling on bacterial pathogenesis,” Nature Reviews Microbiology 2013.

Author Response

Response to the Reviewers of the manuscript Manuscript ID: ijms-1005109

We would like to thank the Reviewer for the time taken to thoroughly read this manuscript, and for the pertinent comments, critics, and suggestions. We are confident that the inputs provided by the Reviewer will help to improve the overall quality and clarity of the final version of the manuscript. We hope that the Reviewer find our responses satisfactory.

Please, find below the Reviewers’ comments in italics and our response following the comment.

The revised manuscript by Makowski is vastly improved. However, several references to LPS structure and modifications  are misquoted and are not relevant to what is stated in text lines 54 to 58 and lines 94 to 100. Please replace citations as suggested below which are more appropriate.

The references for LPS section are not authentic ones. It is best to use well cited papers by experts with relevant coverage. Remove references 14 to 18. Reference 19 is fine. Remove reference 25 not relevant. Also remove reference no 30 as it covers a totally different aspect.

  • Replace ref 14 by most widely used review (Raetz, C.R.H.; Whitfield, C. Lipopolysaccharide endotoxins. Annu. Rev. Biochem. 2002.
  • Replace ref 15 by a very good review by Raetz, C.R.; Reynolds, C.M.; Trent, M. S.; Bishop, R.E. Lipid A modification systems in Gram-negative bacteria. Annu. Rev. Biochem. 2007, 76,
  • Replace Ref 16 by Klein, G et al J. Biol. Chem. 2011, 286,. This deals with lipid A and core alterations.
  • Replace ref 17 by Klein, G.; Müller-Loennies, S …. J. Biol. Chem. 2013, 288 describes phosphate charges and charge compensation by modification of core region.
  • You can cite review by Raetz et al Rev. Biochem. 2007, 76 instead of no 25.
  • Replace ref 30 by B. D. Needham, and M. S. Trent, “Fortifying the barrier: the impact of lipid A remodelling on bacterial pathogenesis,” Nature Reviews Microbiology 2013.

We thank the Reviewer for the input. The references suggested are undoubtedly more fitted to what is referred to in the manuscript. We have thus altered the references accordingly, as suggested.